# The Relationship between Physical Activity and the Objectively-Measured Built Environment in Low- and High-Income South African Communities

**DOI:** 10.3390/ijerph18083853

**Published:** 2021-04-07

**Authors:** Moses Isiagi, Kufre Joseph Okop, Estelle Victoria Lambert

**Affiliations:** 1Research Centre for Health through Physical Activity, Lifestyle and Sport (HPALS), Division of Exercise Science and Sports Medicine (ESSM), FIMS International Collaborating Centre of Sports Medicine, Department of Human Biology, Faculty of Health Sciences, University of Cape Town, Cape Town 7725, South Africa; kufre.okop@uct.ac.za; 2Faculty of Humanities, University of Cape Town, Cape Town 7725, South Africa

**Keywords:** physical activity, built environment, walkability, walking, transportation, recreation, ground-truthing

## Abstract

There is limited data concerning the built environment and physical activity (PA) in a country with a history of sociopolitically motivated, spatial and economic disparities. We explored the extent to which objectively measured attributes of the built environment were associated with self-report or device-measured PA in low- and high-socioeconomic status (SES) communities. Methods: In a convenient sample of residents (*n* = 52, aged 18–65 years) from four urban suburbs in low- and high-income settings near Cape Town, South Africa, self-reported transport- and leisure-time PA, and device-measured moderate-to-vigorous PA (MVPA) data were collected. Built environment constructs derived from individual-level street network measures (1000 m buffer, ArcGIS, 10.51) were obtained. We assessed PA between four groups, based on income and GIS walkability (derived by a median split, low or high SES and low or high walkable). Results: No relationships between self-reported MVPA and GIS-measured walkability were found. Only intersection density was significantly, inversely associated with moderate and total MVPA (rho = −0.29 and rho = −0.31, respectively, *p* < 0.05). In the high SES group, vigorous PA was inversely associated with intersection density (rho = −0.39, *p* < 0.05). Self-report transport PA differed between groups (*p* < 0.013). Conclusions: Results suggest that the construct of walkability may relate to volitional (leisure) and utilitarian (transport) PA differently, in highly inequitable settings.

## 1. Introduction

The burden of physical inactivity is substantial in the country of South Africa [1] as it grapples with a concomitant rise in non-communicable diseases (NCDs), exacerbated by so-called “modifiable risk factors” including physical inactivity, unhealthy eating and overweight, smoking and an excessive alcohol intake [2]. However, there is growing recognition of the ecological, environmental and social determinants of NCDs, which are not under the control of the individual [3]. Economic and spatial inequalities in South Africa have contributed to the disproportionately high burden of NCDs among disadvantaged persons [4] despite social grants and remittances targeting these groups, post 1994 after democracy [5].

The monetary and spatial inequality trends in South Africa may only be addressed through policy interventions and solutions, targeting issues of social and environmental justice [3]. The ecological model provides a framework to understand these interventions. This is based on a premise that healthy behaviors are shaped at both individual and cooperative levels, and factors related to the built and social environments, and public policies [6]. This model has provided a foundational base for examining the correlates and determinants of physical activity and other health behaviors, as they can impact a large population group over time.

One key aspect of this ecological model is the built environment, which may be shaped by various planning and design processes, with the general targets for policies and structural interventions concerning physical activity usually found in the domains of “active travel” and “recreation or leisure time”. This study mainly focused on these two domains. Many studies have documented the relationship between supportive attributes of the built environment and physical activity for transportation and recreation purposes [7], yet most of this evidence is from high-income countries. In Africa, studies of built environment and physical activity relationships remain scarce [8,9] and many questions remain to be answered, including whether the objectively measured attributes of the built environment are associated with either self-reported or device-measured physical activity. Globally, there are a limited number of studies, which have interrogated these relationships [10,11].

In one of the few studies in Africa, Malambo et al. [12] evaluated the relationship between objectively measured attributes of the built environment (using GIS) and physical activity, along with other CVD risk factors (systolic blood pressure, diastolic blood pressure and obesity) in an urban South African context. The study, in general, observed a significant positive relationship between proximity to community centers and shopping centers (500 m) and a lower body mass index and blood pressures, and more physical activity, compared to distances of 1000 m or 1600 m. These results provide supporting evidence that walkable neighborhood environments are associated with lower CVD risk. The study was the first in an African setting to provide evidence of a direct association between objectively measured built environment and physical activity with body mass index, systolic blood pressure and diastolic blood pressure. One limitation of the study area was that it was conducted in a small, geographically defined, low-income community (located near Cape Town), which had a limited variability regarding land use. The study revealed a need for more GIS data to better examined the built environment and physical activity in South Africa. The study also did not establish a link between walkability and physical activity.

The aim of this study, therefore, was to ascertain whether the objectively measured attributes of the built environment were associated with either self-reported or device-measured physical activity in an urban South African setting. Furthermore, we were interested in whether these relationships, should they exist, differ in high- and low-income settings, or by domains of activity.

## 2. Materials and Methods

Materials: A cross-sectional survey design was used to collect information on physical activity and the neighborhood environment from community areas chosen to maximize the variability in neighborhood walkability (low-socioeconomic status (SES)/low walkable, high-SES/low walkable, low-SES/high walkable, high-SES/high walkable).

### 2.1. Setting and Participants

The research study was conducted in four suburbs in the Western Cape Province of South Africa, namely, Langa, Pinelands, Khayelitsha and Table View, and all were in the urban metropole, South Africa. The suburbs were selected using clustered sampling method. Langa and Khayelitsha represented two primarily low socioeconomic status (SES) suburbs, whereas Pinelands and Table View represented suburbs of a higher SES, based on the city development index (CDI) that averaged indices of infrastructure, health, education and income. The suburbs of Khayelitsha and Langa have an average CDI of 0.75 compared to the provincial average of 0.81, whereas the suburbs of Table View and Pinelands have a CDI above 0.90. The four suburbs can be compared, as they are mainly residential, but have various commercial centers with retail, business, recreational facilities and public open space, with varying degrees of land use mix.

This study used a convenience sampling to select 66 adults (aged between 18 and 65 years) from who were regular members of either church or community groups in their respective suburbs for at least three months. Out of this sample, 14 of them had incomplete data, leaving us with a total of 52 adults with complete data needed for our analysis. The 52 included 10 men and 42 women; from Pinelands (*n* = 13), Table View (*n* = 13), Khayelitsha (*n* = 12) and Langa (*n* = 14).

During the recruitment phase, an information letter, including the background and purpose of the study was given to the potential subjects after a visit had been made to community gatherings and the church services within these neighborhoods. Informed, written consent to participate in the study was then obtained.

The sample was further divided into four groups using a median split for the GIS-measured walkability index (in 1000 m buffer) and by the neighborhood income level. *n* = 51 was the total valid and was further divided into (low-SES/low walkable (*n* = 12), high-SES/low walkable (*n* = 13), low-SES/high walkable (*n* = 13) and high-SES/high walkable (*n* = 13)). Approval for the study was granted by the University of Cape Health Sciences Research Ethics committee (HREC REF 293/2016).

#### Dwelling Profile of Participants in Cape Town Study Areas

The four study areas selected in Cape Town metropolis are shown in Figure 1, and a comprehensive description of the these four neighborhoods, and their comparison with other LMIC settings is presented below. The number of households living in the formal dwellings in Cape Town has doubled over the 20 years from 516,867 to 1,032,497 in five years from 2011 to 2016, this indicates an increase of 23%. For the households living in informal dwellings, the report shows that the number increased from 3.3% in 1996 to 6.1% in 2016 with the number of households increasing by 55,859 indicating a 256.5% increase. The population growth in the city of Cape Town is expected to be at 4.2 million people by 2022 and 4.46 million by 2032 [13]. Between 2011 and 2035, 0.6 million more households are expected in Cape Town bringing it to a grand total of 1.7 million households with an average of three people in the households [13].

Compared to the rest of LMICs, Abubakar and Doan [14], show that there is a unique pattern of overcrowded urban core in post-colonial new capital cities in Africa. Cities like Abuja (Nigeria), Gaborone (Botswana), Lilongwe (Malawi) and Dodoma (Tanzania) have failed to provide adequate housing and infrastructure and the projects to improve on these are capital-post colonial cities are exorbitant.

Willemse and Donaldson [15], underscore that the existing park literature in South Africa is limited in scope and dates back to the apartheid era, with barely any information pertaining to community neighborhood park (CNP) use especially in townships. The apartheid’s government policy was based on urban racial segregation and town planning was the prime tool through which new and existing urban landscapes were fashioned. These historical spatial imbalances in the development of residential neighborhoods resulted in the unequal distribution of CNPs, which is clearly seen and portrayed in Cape Town.

Their research sought the perceptions, preferences, needs and uses of CNPs in five black townships in Cape Town including (Khayelitsha, Langa, Gugulethu, Nyanga and Lwandle). Their research revealed that these townships had few CNPs, which therefore entailed travelling greater distances by public transport for access. Furthermore, the lack of private garden space forced the respondents to visit the CNPs and spend more time there thereby participating in either active or passive recreation. The main concerns for the CNPs included safety, maintenance and a lack of CNP facilities.

### 2.2. Measurements

#### 2.2.1. Self-Reported Physical Activity

The international physical activity questionnaire (IPAQ-long) English version was used to measure participants’ self-reported physical activity. This instrument has been previously validated in adults in South Africa with acceptable correlations [16]. Additionally, another study had reported that IPAQ-Short version has a good concurrent validity and test–retest reliability for vigorous-intensity PA, walking, sitting and total PA, and fair construct validity for sitting and moderate PA among Africans [17].

The 31-item long form comprehensively assessed the frequency, duration and intensity of physical activity in the four domains of work, household, transportation and leisure. The IPAQ is used to compute the weekly dose of both moderate and vigorous physical activity (minutes per week = days per week × minutes per usual day during the previous week). We did not translate and back translate the IPAQ into the South African local language spoken in Cape Town. Rather, we had trained local research assistants to administer the survey using local language equivalents of the words to match the intensities.

We specifically examined walking or cycling for transportation and walking and physical activity for recreation. Work-related physical activity was not included, as it is not typically associated with the neighborhood built environment [6]. Therefore, the total min/week in the domains of transport and recreation were summed to estimate overall minutes of self-reported, moderate-to-vigorous physical activity (MVPA) per week. Data that were over-reported were truncated according to the IPAQ protocol (www.ipaq.ki.se, accessed on 10 March 2021).

#### 2.2.2. Device-Measured Physical Activity

Participants were fitted with accelerometers (Actigraph GT3X, Firmware 3:2:1, Actigraph, Pensacola, FL, USA) worn on an elastic belt around their waist. They were asked to wear them for seven days, and only to remove them when bathing or when they went to bed at night. Data were downloaded and analyzed using Actilife 6:10:4 software (ActiGraph, Pensacola, FL, USA).

The Actigraph GTX3 activity monitor was used to assess physical activity objectively. Minute by minute activity counts were accumulated and collapsed into minutes spent at different physical activity intensities, using the Freedson cut-points across the seven days [18]. These intensities included: light (counts less than 759 per minute), moderate (counts between 760 and 5724) and vigorous (counts between 5725 and 9498).

The accelerometer data were collected and aggregated to one-minute epochs. Non-wear time was determined as any period of 60 min or more of consecutive zero counts. Only data from participants with at least 10 h of valid wear time, on at least four days, were included. Counts/minutes were converted into minutes of sedentary time (≤100 counts/min), light, moderate and vigorous-intensity physical activity as previously described [3].

Measured built environment (GIS):

Geographic Information Systems (GIS):

### 2.3. Buffer Size and Type

Using geographic information systems (GISs) (ArcGIS version 10.51), we identified the available physical activity facilities, which included sporting venues, recreational centers and parks within a 1000 m residential buffer. The source of the point data was the City of Cape Town 2011 census. A radial buffer (1000 m) was established [18,19] around the street intersection closest to each participant’s home address. The distance (1000 m) corresponded to a 10–12 min walking time for persons travelling on foot.

The methodology used was based on Adams et al. [20], using the individual-level street network buffer-based GIS measures. The advantage of using this approach was that it placed the participants within the neighborhood and captured destinations that participants could access from the road network. As a result, this method has merits compared to an administrative boundary approach of defining neighborhoods [20].

Walkability measures using GIS:

Walkability:

A walkability index was computed as the sum of Z scores for net residential density, land use mix and diversity and intersection density [20]. The walkability index was adapted from [16] and calculated as:Walkability = ((2 × Z-intersection density) + (Z-net residential density) + Z-retail floor area) + (Z-land use mix)).

This score was adapted from the original measure, which included the retail floor area ratios, which were not available in the present study, but (Adams et al., 2014) [18] captured the two primary theoretical constructs of walkability, proximity and connectivity [21].

Net Residential Density:

Net residential density was computed as the number of dwellings (numerator) divided by the land area dedicated to residential use, within the 1000 m buffer [20]. The residential density was computed for the buffer in the four suburbs as ((dwelling count/residential area) × 1,000,000) (the × 1,000,000)) was used to convert the buffer area from kilometers to meters squared.

Street connectivity:

In this study, [20], street connectivity was operationalized as intersection density. Figure 2 gives and overview of the intersection density of the four urban communities in our study. This was defined as the ratio of the number of intersections within each participant’s buffer (numerator) divided by the total buffer area. Previous papers [22,23] have established intersection density as a measure of route directness, which captures the ability to move to and from destinations in a direct pathway. An intersection in this study was defined as a point where three or more segments intersected after removal of limited-access roads and pseudo intersection nodes [20]. All streets in Cape Town were merged and cut out to fit the buffers in the four suburbs. The street connections were then set to point, and the points were joined in the buffers and aggregated to get each intersection in all the buffers. The intersection density was computed as intersection count/buffer area × 1,000,000.

Land-Use Mix and Diversity:

Four land uses were computed viz. residential, retail-combined, civic/institutional and others. Parcel data was used to quantify the land uses. Land-use mix was calculated using an entropy equation [24] to score the area based on these four land-use types [20].

Statistical Analysis:

Descriptive analysis of participants’ sociodemographic characteristics was undertaken and presented as means ± standard deviation for continuous variables or counts and percentages for categorical variables. The self-reported and device-measured MVPA were also compared between high- and low-SES groups using ANOVA. Between group comparisons for normally distributed were made using independent *t*-tests. Where data were not normally distributed, for device-measured and self-reported physical activity, medians and the lower and upper quartiles were provided in addition to non-parametric Mann–Whitney U tests.

The sample was divided into four groups using a median split for the walkability index (low-SES/low walkable, high-SES/low walkable, low-SES/high walkable and high-SES/high walkable). GIS-measured attributes that comprise the measure of walkability were calculated for the 1000 m buffer zones surrounding the participants’ street address. Spearman’s rho correlations were calculated between the reported physical activity for transport and recreation (IPAQ), device-measured physical activity and GIS measures of walkability. A Kruskal–Wallis test was used to compare self-reported and objectively measured physical activity between the four groups, according to SES and walkability. Statistical tests were considered significant at *p* < 0.05. All data were analyzed using Stat Soft^®^ 2014 version 13 for Windows (IBM Corp: New York, NY, USA).

## 3. Results

Descriptive characteristics of the participants:

The descriptive characteristics of the participants for whom there was complete data (*n* = 52) are presented in Table 1. The participants’ mean age was 41.4 ± 12.7 years in the lower-SES groups and 45.8 ± 12.7 years in the higher-SES groups. Significant differences were found for BMI (*p* < 0.05) between the two-SES groups. Of those participants in the high-SES neighborhoods, 88% were either married or living with a partner, compared to only 19% from the low-SES communities. Additionally, 92% of persons surveyed in the high-SES suburbs compared to 44% from low-SES neighborhoods had access to at least one private motor vehicle.

There were no significant differences between the two SES groups for device-measured light and moderate physical activity. However, time spent in vigorous physical activity was significantly higher in the high-SES groups. For self-reported physical activity, the median time participants reported for TPA in the combined-SES was 60 min/wk with 25% reporting ≤ 10 min/wk of transport-related physical activity.

Self-reported leisure-time physical activity in the combined-SES (LPA) and transport physical activity (TPA) and total physical activity (TTL PA) were statistically significantly different. These differences are due to chance or lack of statistical power.

Self-reported and device-measured physical activity and GIS walkability:

There was no relationship between any self-reported physical activity and GIS measured attributes of walkability in 1000 m buffer as shown in Table 2. Data on objectively measured (GIS) walkability and device-measured physical activity and SES categories (low and high) were compared for 1000 m buffers, using Spearman’s rho coefficient, and presented in Table 3. For the high SES group, intersection density in the 1000 m buffer was inversely associated with device-measured vigorous physical activity only (*p* < 0.05). When groups were combined, the inverse relationship between intersection density and physical activity persisted. In the 1000 m buffer, intersection density was inversely associated with both moderate (r = 0.29, *p* < 0.05) and moderate-to-vigorous physical activity (r = 0.31, *p* < 0.05).

Self-report and objectively-measured physical activity (MPVA) and GIS walkability when apportioned by income:

There was a significant overall difference in self-reported physical activity in the domain of transport (*p* = 0.036) with significant between group differences between high-SES/low walkable vs. low-SES/high walkable (*p* = 0.013) as reported in Table 4. Residents in the low-SES/high walkable neighborhoods reported more transport-related physical activity compared to high-SES/low walkable. There was a significant overall difference in device-measured vigorous physical activity between income groups (*p* = 0.016), with between group differences for the low-SES/low walkable vs. high-SES/low walkable groups (*p* < 0.04).

## 4. Discussion

This study examined the differences in self-reported physical activity (transport and leisure domains) and device-measured physical activity (MVPA), in groups apportioned according to income and GIS measured walkability within a 1000 m buffer. In general, there was no relationship between self-reported physical activity and walkability (or the components of walkability, measured using GIS), irrespective of income level. Conversely, the device-measured physical activity for all groups was inversely associated with intersection density. There were also observed differences in transport-related, self-report physical activity between SES groups.

Previous studies have established that adults walk more for transportation in walkable neighborhoods [25]. However, in the current study, we found no such association, in either income group, irrespective of the GIS-measured walkability. This finding is important and is similar to those from Sallis et al. and Thorton et al. [26,27] summarized in a review by Adkins et al. [28]. These researchers suggest that low-income/SES respondents may not experience all the benefits of living in a walkable neighborhood unless other needs are met. In the low SES groups, transport-related physical activity was significantly higher, and in the high-income groups, car ownership was significantly higher. Thus, it suggests that transport physical activity in the low income groups was not a matter of choice, and that in the high-income groups, it was not a necessity, These results highlight the differences in utilitarian physical activity (such as walking for transport) and leisure time or volitional activity, based on choice.

A vast majority of the literature has focused on examining associations between aspects of the built environment and modes of choice [29,30,31]. These studies generally support the relationship between physical activity and the physical environment based on three key elements of the physical environment: greater proximity to retail destinations [32,33,34], high connectivity [35,36] and land use mix [23,37]. The current study examined the associations between four elements, including land use mix, intersection density, residential density and transport density and highlighted that intersection density had significant, but inverse associations with physical activity. Therefore, in this study settings, walkability may involve the creation of less intersection density and more space [38]. The modification of the built environment to create “walkable built environments” in low-SES communities, with high density and poor infrastructures, may require a better understanding of the user needs and expectations, and an upstream influencing factor that may impact more broadly, on personal safety [39]. This may help to reduce the SES inequalities concerning participation in physical activity, while embedding solutions within the “lived experience” of the community.

Based on GIS and other measures, not all socioeconomic groups may benefit equally from the current notion of “walkable” built environments [26]. Studies that have examined the influence of the built environment on active transport for different socioeconomic groups have obtained mixed results [26,27,40,41]. This suggests that the equivocal nature of these findings could be due, in part, to the way the built environment measures were determined. They point out that studies that examined the built environment in residential settings and ignore non-residential destinations could explain some of these differences. They also suggest that built environment features along the entirety of the spatial trajectory, from origin and destination, may have the potential to influence active transport mode of choice. In their study, they again show that the built environment has a weaker association with the active transportation of those from low-SES neighborhoods.

This study also points to socioeconomic inequalities, cultural and contextual differences in the environment (i.e., crime rates, poor access to physical activity facilities and public transportation), which are common to low- and middle-income countries, where PA is used for utilitarian, rather than recreational purposes. There is also the issue of environmental justice. The effect of environmental and social injustice still plagues South Africa post-1994. The skewed spatial legacies are indicated to determining opportunities in South Africa until now [4]. For instance, it is commonly observed that urban parks in South Africa appears to be inequitably distributed within cities, especially with communities of lower SES and people of color having inferior geographic access to parks thereby constraining the frequency of park use [42]. The possible confounding factors such as inequalities (racial, spatial, racial and monetary), and poverty [4,5] were not considered within the scope of our study, and this might have some effects on our results.

Strengths and limitations:

A strength of the present study was in its design to recruit participants from four different neighborhoods, based on income or socioeconomic status and residential density in an urban South African setting. The other strength included the use of accelerometers and GIS to objectively assess PA, and the perceived and objectively-measures of built environment. This study is one of the first to do so in Africa.

The study has fundamental limitations to note. Conducting the study among residents of one neighborhood type or as part of a convenience sample (as participants in one area attend the same church within the same community) may restrict environmental variability. Restricted variability could, in turn underestimate the strengths of environmental–physical activity associations in environmental studies [6,20,26,41]. The small sample size and its possible effect on the statistical power of the study is one main weakness. This could reduce the ability to detect true differences, and may limit the generalizability of the study to other urban settings both in South Africa and more broadly in Africa.

## 5. Conclusions and Recommendations

The results of this study showed a mismatch between physical activity and the objectively measured attributes of the built environment that have been previously associated with “walkability”, largely in high-come countries. The findings are significant, especially given the high social disparities that face South Africa. This study suggests that persons living in low-income settings may not experience all the benefits of living in a walkable neighborhood, particularly if physical activity that is undertaken in utilitarian and not volitional. There is a need to consider the constraints to physical activity in low SES groups, along with the built environment, in order to address social and environmental justice in the promotion of physical activity, at a population level.

The findings of this study could be relevant for health promotion and policy advocacy in sub-Saharan Africa, particularly in settings with a history of social and politically motivated, spatial disparities and inequities. Understanding the environmental correlates may lead to better strategies to promote physical activity in the low-income setting. Further evaluation of the relationship between physical activity and the built environment is needed in the broader South African population to strengthen evidence-based recommendations for creating better and safer communities with the built environment that improve access to safe and enjoyable opportunities in the African region.

## Figures and Tables

**Figure 1 ijerph-18-03853-f001:**
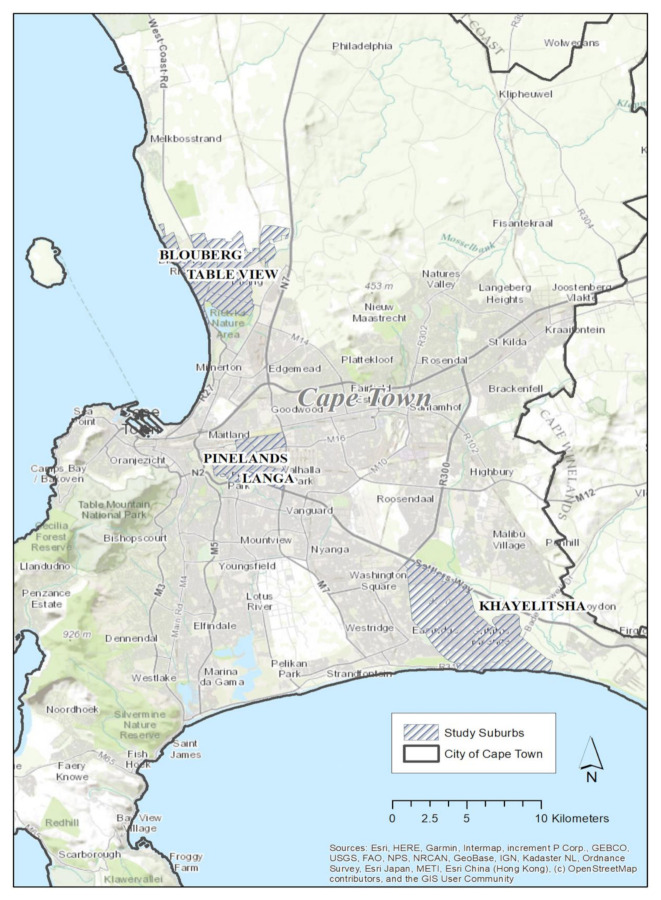
The map of visual representation of Cape Town, with the four study areas.

**Figure 2 ijerph-18-03853-f002:**
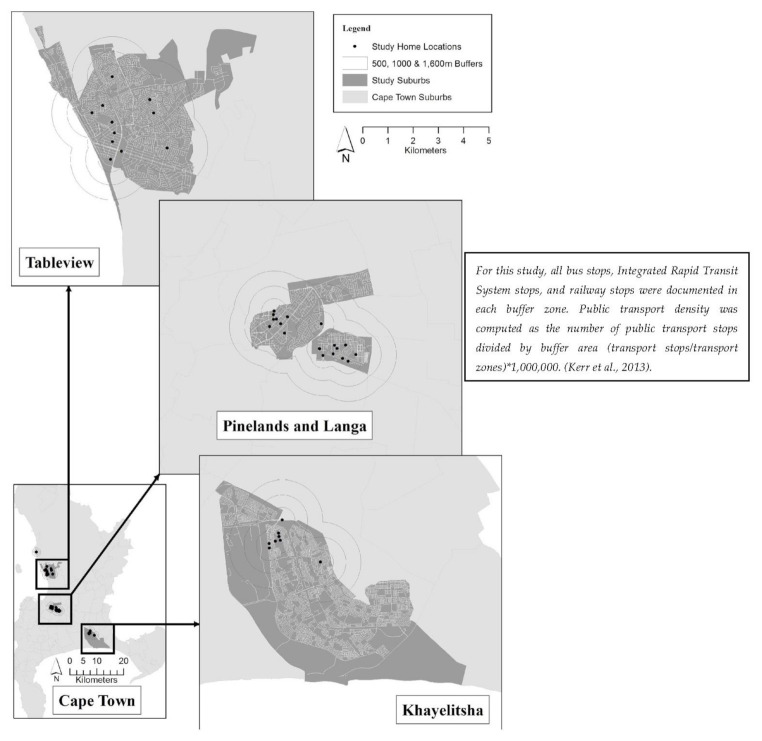
An overview of intersection density in the four study suburbs.

**Table 1 ijerph-18-03853-t001:** Descriptive statistics for participants’ demographics SES group.

Variables	Low SES (*n* = 26)	High SES (*n* = 26)	Combined SES (*n* = 52)	*p* Value ^+^
**Age ^a^**	45.77 (10.8)	41.38 (12.7)	43.58 (11.9)	0.19
**BMI ^a^**	28.9 (23.9)	33.9 (23.0)	31.1 (23.34)	0.05 **
**Marital Status (*n*, %)**				
Married/Living with partner	5 (19.2)	23 (88.4)	26 (50.0)	0.001 **
Single	16 (64)	2 (7.6)	19 (36.5)	
Widowed	4 (16)	1(3.8)	5 (9.6)	
**Level of Education (*n*, %)**				
Completed High school	3 (12)	3 (11.5)	6 (11.5)	0.001 **
Diploma/Higher Diploma	5 (20)	14 (53.8)	19 (36.5)	
Bachelor’s degree	1 (4)	3 (11.5)	4 (7.7)	
Graduate degree	0 (0)	6 (23.1)	6 (11.5)	
Completed High school	3 (12)	3 (11.5)	6 (11.5)	
**Motor Vehicle Use**				
None	11(42.3)	2 (7.6)	14 (53.8)	0.001 **
One or more	15 (57.3)	24 (92.4)	12 (46.2)	
**Device-Measured Physical Activity ^b^**				
MPA	232 (11; 353)	185 (130; 273)	209 (118; 327)	0.78
VPA	0 (0; 2)	4 (0; 11)	1 (0; 5)	0.01 **
MVPA	232 (114; 354)	195 (136; 274)	216 (120; 327)	0.99
**Self-Reported Physical Activity ^b^**				
TPA	113 (45; 180)	25 (0; 70)	60 (10; 180)	0.78
LTPA	113 (30; 180)	180 (100; 180)	151 (41; 180)	0.99
TTL PA	403 (255; 540)	408 (240; 450)	403 (248; 525)	0.69

^+^ Values based on independent *t*-tests for continuous variables, ** statistically significant difference by gender (*p* < 0.05), **^a^** values for age and BMI (body mass index) are mean ± standard deviation; **^b^** device measured and self-reported physical activity because the physical activity outcomes were skewed, the median and interquartile range (25th and 75th percentile values) were tabled. PA are measured in minutes/week. SES: Socioeconomic status; MPA: Moderate physical activity; VPA: Vigorous physical activity; MVPA: Moderate-to-vigorous physical activity; TPA: Transport physical activity; LPA: Leisure-time physical activity; TTL PA: Total physical activity.

**Table 2 ijerph-18-03853-t002:** GIS-measured walkability and self-report physical activity in the 1000 m buffers.

SES	Low-SES			High-SES			Combined-SES		
**Self-reported physical activity domains** **→**	Transport physical activity truncated	Leisure physical activity Truncated	Total PA Truncated	Transport physical activity truncated	Leisure physical activity Truncated	Total PA Truncated	Transport physical activity truncated	Leisure physical activity Truncated	Total PA Truncated
**GIS–Measured Walkability and Self-Report Physical Activity in the 1000 m Buffer**									
**r (Spearman’s rho) ^a^**									
**GIS Measured ↓**									
Overall Walkability	0.22	0.39	0.22	0.05	−0.12	−0.15	0.07	0.16	0.05
Land use Mix	0.07	0.20	0.14	−0.11	−0.14	−0.08	−0.13	0.06	−0.03
Intersection Density	0.10	0.22	0.17	−0.03	−0.28	−0.31	−0.01	0.03	−0.04
Residential Density	0.01	−0.14	−0.19	−0.09	−0.19	−0.16	0.11	−0.26	−0.12
Transport Density	0.26	0.32	0.12	0.02	−0.06	−0.11	0.05	0.10	−0.01

**^a^** Spearman’s (rho) correlations tests were used to present comparisons between GIS-measured walkability and self-report physical activity in the 1000 m buffers.

**Table 3 ijerph-18-03853-t003:** GIS-measured walkability and device-measured physical activity in the 1000 m buffers.

SES	Low-SES			High–SES			Combined-SES		
Device Measured Physical Activity (min/wk) →	Moderate	Vigorous	Total MVPA	Moderate	Vigorous	Total MVPA	Moderate	Vigorous	Total MVPA
GIS–Measured Walkability and Device-Measured Physical Activity in the 1000 m Buffers									
	r (Spearman’s rho) ^a^								
**GIS Measured** **↓**									
Overall Walkability	−0.13	0.14	−0.13	−0.09	−0.10	−0.10	−0.13	0.04	−0.13
Land use Mix	0.03	0.22	0.03	−0.08	0.07	−0.09	0.05	0.24	0.04
Intersection Density	−0.30	−0.16	−0.31	−0.09	−0.39 **	−0.19	−0.29 **	−0.20	−0.31 **
Residential Density	−0.15	−0.09	−0.15	−0.21	0.04	−0.16	−0.18	−0.10	−0.18
Transport Density	−0.09	0.27	−0.08	−0.14	0.09	−0.11	−0.10	0.18	−0.09

Spearman’s (rho) **^a^** correlations tests were used to present comparisons between GIS-measured walkability and objective physical activity in the 1000 m buffer. ** Comparisons that were statistically significant at *p* < 0.05.

**Table 4 ijerph-18-03853-t004:** Comparisons of self-reported and device-measured physical activity by SES-walkability status (low-high SES and low-high walkability) **^a^**.

NEWS 1 Sub-Scale	Low-SES/ Low w *	High-SES/ Low w *	Low-SES/ High w *	High-SES/ High w *	*p* Values **
**SES-Walkability Group (SES-W) Categories** **→**	**1**	**2**	**3**	**4**	
	Median (lower and upper interquartile range) ^Y^				
**Self-Reported Physical Activity (IPAQ)**					
**Transport Physical Activity (min/wk)**	75 (30; 120)	20 (00; 70)	180 (45; 180)	30 (0.0; 75)	Overall *p* = 0.013 ** 2 vs. 3 = 0.024 **
**Recreation Physical Activity (min/wk)**	100 (20; 180)	180 (120; 180)	120 (60; 180)	180 (100; 180)	Overall *p* = 0.24
Median (lower and upper interquartile range) ^Y^					
**Device-Measure (Accelerometer)**					
**Vigorous Physical Activity** **(min/wk)**	0.0 (0.0; 1.0)	4.0 (2.0; 11)	0.0 (0.0; 7.0)	1.0 (0.0; 5.0)	Overall *p* = 0.034 ** 1 vs. 2 = 0.039 **
**Moderate to Vigorous Activity** **(min/wk)**	264 (114; 354)	239 (169; 323)	155 (67; 383)	164 (136; 271)	Overall *p* = 0.639

**^a^** Kruskal–Wallis tests were used to present the multiple comparisons between self-reported physical activity, measured physical activity and SES-W. * w—walkability. ** *p*-values are reported for only comparisons that were statistically significant—*p*-value < 0.05 (N/B: 1 vs. 4—means low-SES-low walkability category vs. high SES-high walkability categories). ^Y^ Figures presented on the table are based on median, lower and upper interquartile ranges.

## Data Availability

Not applicable.

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
