# Peer review of "The Relationship between Physical Activity and the Objectively-Measured Built Environment in Low- and High-Income South African Communities"

_ijerph, 2021, doi:10.3390/ijerph18083853_

Round 1

Reviewer 1 Report

General comments:  This is an elegantly designed study to examine the associations between environmental supports (pedestrian/cycling infrastructure) and transportation and leisure-time moderate to vigorous physical activity among low SES and higher SES populations living in the greater Cape Town SA suburbs.  The methods are well described in assessing the built environments (GIS) and physical activity levels (IPAQ and wearable devices).  Statistical/analytic methods are appropriate for such data and results presented support the narrative.  The discussion is appropriate and stays within the bounds of the data.  Conclusions are consistent with the data, but rather strong in light of the limitations identified by the authors.  

Such limitations need further discussion with a tempering of the findings from this study by responding to the following:

1) The subject sampling and sample size (power) of the study is a primary limitation of the current study.  It is unclear whether subjects are truly representative of the communities/neighborhoods being examined.

2) The GIS measures are strong and the handling of such environmental data is to be commended.  However, further examination of the presence or absence of public transport should be conducted in both low- and higher-SES.  Since active transport in lower SES communities has been shown to be strongly associated with resident physical activity.  

3) A further discussion about the potential determinants of active transport and leisure-time physical activity among lower SES populations and the cultural environments associated with such behaviors needs to be provided (See Ford ES, et al. American Journal of Epidemiology (older citation) as well as the US Preventive Services Task Force's Guide to Community Preventive Services - Physical Activity)

4) In light of the above, the title of the manuscript should indicate the hypothesis generating value of the study and be deemed a 'pilot study' with the recognition that further examination of these findings be part of a much larger and population-based study of these correlates.  

Author Response

Thank you so much for the comments -Please see the attachment with the responses

Reviewer 2 Report

The paper is well-structured.  Methods were well-written and results were connected to review of literature. I think that the authors did a great job in writing this paper!

I conducted a major study here in the US but my n value was over 4000.  Because this is the first study to be conducted in South Africa, I find that research methods that the authors conducted are good.  They do acknowledge the limited sample (52) and limited number of neighborhoods (4).  They also used correlations rather than regressions. I visited their statistics again and the choice of statistis are correct.

Strengths: well-designed study given the fact that obtaining data is not easy and is not readily available when compared to U.S. 2. Recommendations to add to manuscript:

  • The authors acknowledge the limited sample size already.  I would recommend that they mention that this is a pilot study (somewhere in the abstract and in the introduction) and add directions for future research on how they intend to build on their study. 
  • For readers who are not familiar with South African neighborhoods, I would recommend the authors to add a description of the 4 neighborhoods (and what factors did they control for).  I would give an emphasis also on transportation modes (most studies in the U.S. are easy to relate to because of familiarity of modes of transportation). Are these neighborhoods automobile reliant, what about the public transportation system, is it affordable?!  This description will add to the context to the study.

Author Response

Dear Reviwer

Thank you so much for the comments.  We appreciate-

Do find attached the responses

Thank you again

Round 2

Reviewer 1 Report

This revised manuscript is improved and provides the reader with a clear understanding of the methods and approach used to explore this important question addressing perceived and real access to environmental supports for physical activity by SES.  A strength of this study is the use of both objectively measured levels and self-reported levels of physical activity among study subjects along with the geographic and demographic contextual characteristics of the lower SES and higher SES communities.  The authors seek to address the topic of equity in terms of access to opportunities for physical activity, with a marked significance in participation in vigorous levels of physical activity when comparing lower SES with higher SES subjects.  Similar findings have historically been found to exist in the United States as well (See Ford ES, et al.,  American Journal of Epidemiology 1991; 133:1246-56).  The only concern is the small sample size which limits statistical power when stratifying results.  The authors appropriately make note of this in the discussion section.  However, perhaps a slight change in title to include either the term: a pilot study or preliminary findings might be in order.  The study provides a solid hypothesis generating purpose, but as with many cross sectional studies provides little in determining the direction of these associations and limits generalizability of the findings.

Author Response

The response are provided in the attach MS word file.

This manuscript is a resubmission of an earlier submission. The following is a list of the peer review reports and author responses from that submission.

Round 1

Reviewer 1 Report

This is well written piece of work. I have few comments to offer:

The authors can use the acronym PA for physical activity

PA related to work was not measured since it was not considered important. A reference is needed to support this statement

I imagine that the authors used the English version of the IPAQ. They should provide information regarding its psychometric properties, specially in African population.

Lines 198-200 can be erased, as well as 261 to 264

The design of the study provides and opportunity for comparing self-perceived (IPAQ) vs objectively (accelerometer) PA. My advise is to provide information in this regard, and compare low vs high SES groups. Which one self-perceived better the amount of PA performed?

Reviewer 2 Report

It is overall very well presented the research study of objectively measured built environment and PA in different SES in South African, very interesting to read.

Minor suggestions are

1) to rich the introduction to talk more about the SES in South African, how big is the difference between the high and low, because it is so different in different counties and it may reveal other confounding factors that may contribute to the NCDs and make sure you mention those in discussing the associations.

2) the authors, please consider talk about the study power, because of small sample size that you selected and grouped into 4 categories, it is hard to not doubt the power.

3) when making the conclusion, make sure that you discuss the confounding factors that you did not take into consideration in the study, and address the small sample size and unique geographic differences.